# Therapeutic Drug Monitoring and Biomarkers; towards Better Dosing of Antimicrobial Therapy

**DOI:** 10.3390/pharmaceutics16050677

**Published:** 2024-05-17

**Authors:** Eman Wehbe, Asad E. Patanwala, Christine Y. Lu, Hannah Yejin Kim, Sophie L. Stocker, Jan-Willem C. Alffenaar

**Affiliations:** 1Faculty of Medicine and Health, School of Pharmacy, The University of Sydney, Sydney, NSW 2006, Australia; eweh7708@uni.sydney.edu.au (E.W.); asad.patanwala@sydney.edu.au (A.E.P.); christine.lu@sydney.edu.au (C.Y.L.); hannah.kim@sydney.edu.au (H.Y.K.); sophie.stocker@sydney.edu.au (S.L.S.); 2Department of Pharmacy, Westmead Hospital, Sydney, NSW 2145, Australia; 3Department of Pharmacy, Royal Prince Alfred Hospital, Sydney, NSW 2050, Australia; 4Department of Pharmacy, Royal North Shore Hospital, Sydney, NSW 2065, Australia; 5Kolling Institute, Faculty of Medicine and Health, The University of Sydney, The Northern Sydney Local Health District, Sydney, NSW 2065, Australia; 6Sydney Institute for Infectious Diseases, The University of Sydney, Sydney, NSW 2145, Australia; 7Department of Clinical Pharmacology and Toxicology, St. Vincent’s Hospital, Sydney, NSW 2010, Australia

**Keywords:** therapeutic drug monitoring, biomarkers, precision medicine, PK/PD

## Abstract

Due to variability in pharmacokinetics and pharmacodynamics, clinical outcomes of antimicrobial drug therapy vary between patients. As such, personalised medication management, considering both pharmacokinetics and pharmacodynamics, is a growing concept of interest in the field of infectious diseases. Therapeutic drug monitoring is used to adjust and individualise drug regimens until predefined pharmacokinetic exposure targets are achieved. Minimum inhibitory concentration (drug susceptibility) is the best available pharmacodynamic parameter but is associated with many limitations. Identification of other pharmacodynamic parameters is necessary. Repurposing diagnostic biomarkers as pharmacodynamic parameters to evaluate treatment response is attractive. When combined with therapeutic drug monitoring, it could facilitate making more informed dosing decisions. We believe the approach has potential and justifies further research.

## 1. Introduction

Early and optimal antimicrobial treatment is critical, especially for immunocompromised patients with life-threatening infections. Optimal antimicrobial treatment refers to the attainment of a successful therapeutic response at the lowest dose possible to limit toxicity while preventing the development of drug resistance [1]. This requires consideration of not only the pharmacokinetics of the drug but also the pharmacodynamics.

Therapeutic drug monitoring (TDM) is a useful method used in clinical practice to assess the pharmacokinetics of a drug in an individual patient and tailor the drug dose to achieve the desired drug exposure. Blood samples immediately before the administration of a new dose are usually collected, reflecting the minimum drug concentration (C_min_ or trough concentration) [2]. 

The minimum inhibitory concentration (MIC) is traditionally used as a pharmacodynamic parameter. It involves the quantification of the minimum antimicrobial drug concentration needed to inhibit pathogen growth and determines whether a standard dose is suitable (i.e., the pathogen is susceptible), an increased dose is required for effective treatment (intermediate) or a switch to another antimicrobial drug should be made. However, there are several issues with using MIC measurements from patient specimens. Firstly, MIC is calculated using in vitro assays and does not account for physiological impacts on concentrations in the human body (e.g., drug penetration) [3]. Secondly, MIC results are assay-dependent, and results may therefore vary (1 dilution step) [3]. Finally, it takes time to determine the MIC. While species identification and molecular susceptibility test results can be obtained relatively quickly, culturing pathogens to determine an MIC could take weeks for some microbials (e.g., tuberculosis and fungus) [4,5]. For critically ill patients, this turnaround time is too long to be helpful. In addition, for some microbial infections, especially invasive fungal infections, obtaining a sample to test in the first place is challenging due to the risk associated with taking a biopsy from the infected tissue [6]. Ultimately, there is a need to identify other pharmacodynamic markers to determine the response to antimicrobial therapy.

Herein, we describe the potential for diagnostic biomarkers [7] to be repurposed as pharmacodynamic biomarkers to monitor antimicrobial treatment responses in conjunction with TDM. We chose specific microbial infections, including invasive aspergillosis, invasive candidiasis, tuberculosis, cytomegalovirus and sepsis, to show the breadth of biomarkers and how they can be utilised. 

## 2. Invasive Aspergillosis and Galactomannan

*Aspergillus*, an opportunistic fungal species, is the most common cause of invasive mould infections in immunocompromised and critically ill patients [8,9,10]. Invasive aspergillosis has a high mortality rate (38–79%), and thus, rapid and optimal antifungal therapy with first-line azole antifungals is critical [11]. TDM is recommended to guide the dosage of azole antifungals—voriconazole and posaconazole, and to a lesser extent, isavuconazole—due to their narrow therapeutic ranges and the high interpatient variability in pharmacokinetics [12]. However, current TDM practice focuses more on pharmacokinetics than pharmacodynamics. 

Galactomannan, a fungal polysaccharide, is synthesised by the *Aspergillus* species and can be detected in the serum and bronchoalveolar lavage fluid of infected patients [13]. While galactomannan is used for the diagnosis of invasive aspergillosis in clinical practice [14], there is emerging evidence showing its potential as a pharmacodynamic biomarker to monitor response to treatment as well. 

Early changes in galactomannan levels were associated with treatment response in patients with invasive aspergillosis [15,16]. Two studies investigated galactomannan levels collected in patients that previously participated in two separate randomised controlled trials (*n* = 71, *n* = 114, respectively). They suggest that differences in galactomannan levels at baseline and after a week of therapy could be an early indicator of patients who have had an initial response to antifungal therapy and will likely experience treatment success. One study found that after a week of antifungal therapy, a reduction of galactomannan value by >35% from baseline meant a patient was 3.8 times more likely to have a complete/partial response to therapy at 12 weeks [15]. The other deduced that a 1 unit decrease in galactomannan level (from baseline) by day 7 of therapy doubled the chance of a partial/complete response to therapy at 12 weeks (odds ratio, 2.2; 95% CI, 1.2–4.0) [16]. 

Considering galactomannan levels in conjunction with TDM is more informative than interpretation in the absence of information on drug exposure. For example, an increase in galactomannan from baseline could potentially indicate a poor antifungal response, and in combination with a low-normal drug concentration, it could trigger dose escalation or switching of antifungal therapy in cases where drug concentrations are already close to the threshold of tolerability. On the other hand, a reduction in galactomannan despite low drug exposure could indicate that the current dose is effective and no dose escalation is necessary. This is important because antifungals are associated with severe adverse effects [17], and initiating a potentially unnecessary dose escalation may increase the risk of dose-related adverse effects. 

The fact that several in vitro and in vivo studies have identified that reductions in galactomannan are dose-/concentration-dependent in different strains of *Aspergillus* and across antifungals including voriconazole, posaconazole, isavuconazole and amphotericin B affirms the usefulness of this biomarker with TDM [18,19,20,21,22]. A persistent lack of decline of galactomannan despite increases in antifungal doses could indicate that the fungi might be resistant or that the drug does not reach the site of infection [19]. 

Several pharmacokinetic-pharmacodynamic (PK-PD) models have been developed to calculate target exposure values [16,19,20,21,22,23]. While a ratio of AUC to MIC is traditionally used for these models, obtaining an MIC is difficult for patients with invasive aspergillosis because the pathogen is seldom recovered and clinicians rely on galactomannan positivity, imaging and host factors to start treatment [14,16,22,23]. Therefore, a novel ratio of AUC to the concentration of antifungal that induces a half-maximal reduction in galactomannan levels was used by Huurneman et al. (2016) to model the PK-PD index of voriconazole in children. Despite the small data set (*n* = 12), the model had a good fit, and patients with a ratio less than 6 tended to have lower final galactomannan levels (*p* = 0.07) [23]. However, a randomised controlled trial is needed to determine the value of this PK-PD index compared to a therapeutic range based only on drug concentrations [23].

## 3. Invasive Candidiasis and 1,3-B-D-Glucan

*Candida*, another opportunistic fungal species, is the most common cause of invasive yeast infections. Like invasive aspergillosis, invasive candidiasis is a systemic infection associated with a poor prognosis and high mortality rates (30–60%) [24,25]. However, since galactomannan is mostly specific for the *Aspergillus* species, it is not used as a biomarker for invasive candidiasis [26]. 1,3-β-D-glucan (BDG), a cell wall polysaccharide component of most fungi, is a promising biomarker for the diagnosis of invasive candidiasis [27]. BDG is released into the bloodstream during tissue invasion [28]. Compared to traditional methods like fungal cultures, BDG testing is typically performed on blood samples, and results can be reported within hours.

BDG testing has a sensitivity of 74–86% for the diagnosis of invasive fungal infections [29]. BDG levels typically rise during an invasive fungal infection and decrease after the commencement of antifungal therapy. The frequency of BDG testing during treatment depends on the individual patient and disease severity, where more frequent monitoring may be helpful in those with more severe disease. 

The multicentre CandiSep randomised controlled trial compared BDG-guided early antifungal therapy in sepsis patients at risk of invasive candidiasis with culture-based targeted therapy (standard care) [30]. Two BDG samples were taken for each patient (one hour after randomisation and 24 h after enrolment, respectively), and a BDG concentration of ≥80 pg/mL was considered a positive result and led to the initiation of antifungal therapy. The study did not demonstrate any survival benefit from the implementation of two positive BDG results for early invasive candidiasis diagnosis and treatment. Further, in the BDG-guided arm, antifungal therapy was initiated in almost half of patients, despite the low rate of invasive candidiasis (14%). Therefore, in critically ill patients carrying a low risk of invasive candidiasis, a BDG-guided pre-emptive approach cannot be recommended due to the high likelihood of antifungal overuse in the absence of *Candida* infection. It is important to note that the CandiSep study’s limitations included protocol deviations and a lower prevalence of invasive candidiasis than expected. 

Clearly, the use of BDG as a standalone biomarker has limitations, including its poor specificity (60%) and low positive predictive value (<15%) in patients with low-intermediate invasive candidiasis risk [29]. BDG can be elevated in non-fungal conditions like bacterial infections, inflammatory bowel disease and exposure to some medications, which can lead to false-positive results. Low fungal burden and technical issues with the assay may give false negative results. Repeating the test and/or increasing the cut-off value can help increase specificity and the positive predictive value [31]. While BDG may not be the optimal approach for guiding the initiation of antifungal treatment in critically ill patients, its use in guiding the discontinuation of therapy shows promise. Recent trials support the use of a biomarkers-driven strategy (using BDG alone or in combination with mannan/anti-mannan antibodies) as a rule-out diagnostic tool, allowing prompt and safe discontinuation of empirical antifungal therapy in patients without mycological confirmation of invasive yeast infection [32,33]. This strategy has been included in current guidelines and antifungal stewardship programmes [34]. 

When considering BDG as a pharmacodynamic indicator, serial measurements could track the effectiveness of treatment and guide dose adjustments or treatment duration in certain patient populations. While the sensitivity/specificity of BDG are low, serial BDG testing in patients with haematologic malignancies and those who have undergone an allogeneic hematopoietic stem cell transplant led to higher specificities (76–99%) and negative predictive values (87–96%) [27]. Persistent or rising BDG levels, despite normal antifungal exposure, may indicate treatment failure or fungal resistance. This information can prompt clinicians to switch antifungal agents or implement other therapeutic strategies. Decreasing BDG levels can help guide decisions on antifungal therapy duration, potentially shortening treatment courses in patients with a good clinical response. Cut-off values for BDG levels indicating successful treatment are not yet defined and may vary depending on the fungal species, underlying condition and antifungal agents used [35]. Therefore, sequential monitoring (every 24–48 h), taking into account the half-life of BDG, can indicate trends in response to therapy [36]. Research suggests that a decrease in BDG level by 50% or more from baseline after one week of antifungal therapy may be indicative of successful treatment [35]. 

## 4. Cytomegalovirus and Viral Load

Cytomegalovirus (CMV) is a highly prevalent viral infection [37]. While CMV is not usually a severe infection in an otherwise healthy person, it is associated with high morbidity and mortality in immunocompromised patients [38,39]. As such, optimising antiviral therapy is important for these patients. Currently, viral load, the quantification of virus DNA in an infected patient, is used to diagnose CMV and monitor response to antiviral drug therapy [40,41]. 

The clinical utility of viral load in the monitoring of antiviral treatment responses is well established. Viral load monitoring is recommended on a weekly basis because CMV DNA has a half-life of 3–8 days, meaning any earlier tests should be interpreted carefully, especially those showing a lack of decline in viral load [41,42]. Administration of antiviral therapy, namely ganciclovir, is associated with a rapid decrease in viral load [43,44]. Compared to baseline, a consistent or increased viral load could potentially indicate subtherapeutic antiviral concentrations or treatment failure [41,45]. A persistently elevated viral load could imply inadequate dosing or drug resistance and the need for alternative antiviral therapy [41,46,47]. Importantly, to avoid misinterpretation of viral load trends, the type of assay and sample used should remain consistent [48]. An assessment of treatment response should be made after 2 weeks of antiviral therapy. Treatment is ceased when the viral load becomes undetectable, preferably for two consecutive weeks [41].

Routinely monitoring ganciclovir drug concentrations, the first-line treatment of CMV, is currently not recommended in clinical practice because there is not enough strong evidence to support its application [41]. A recent retrospective study showed no association between the attainment of predefined target trough ganciclovir concentrations and clinical response or drug safety [49]. This is despite the need to optimise therapy given ganciclovir’s toxicity and the high interpatient variability in exposure [50].

Lack of knowledge of the ganciclovir exposure target may explain why studies, to date, have been unable to demonstrate a benefit of TDM. The current therapeutic range is based on expert opinion or estimates from the IC_50_ of ganciclovir [49,51]. Different ranges have been used, making it difficult to compare results across studies. Furthermore, the IC_50_ of ganciclovir refers to the inhibitory concentration required to reduce viral replication by 50%, which is more of a laboratory measure as its relevance for clinical use is limited because it does not make sense to target only 50% inhibition of viral replication in patients. Moreover, ganciclovir only exhibits antiviral activity after intracellular phosphorylation and activation by a viral kinase [50]. The extracellular unphosphorylated ganciclovir measured in plasma might not reflect intracellular levels and thus fails to accurately describe IC_50_. Secondly, one study suggests that using overall drug exposure—area under the concentration time curve over 24 h (AUC_24_)—for TDM rather than trough concentrations may predict response to treatment better. This reflects either a poor correlation between C_min_ and AUC or the fact that AUC/MICs predict response better than T > MIC (i.e., time the concentration of antiviral is greater than the MIC) [51]. 

## 5. Tuberculosis and Interferon-Inducible Protein 10

Tuberculosis (TB) is the second leading cause of death from a single infectious source after COVID-19, and multi-drug resistance is an ongoing issue in the treatment of TB [52]. 

Interferon-inducible protein 10 (IP-10) has emerged as a promising biomarker in the diagnosis and monitoring of TB. IP-10 plays a crucial role in attracting T-helper type 1 lymphocytes to sites of infection, making it a valuable indicator of TB infection [53].

However, the use of IP-10 to differentiate between active TB and latent TB has yielded inconsistent results. Some studies report elevated IP-10 levels in active TB [54,55,56], while others report decreased levels when compared to latent TB patients [57,58]. This variability likely stems from differences in assay methods, as well as variations in the immune states of TB infections [53]. More recent data report significantly elevated IP-10 at both protein and gene levels in active TB patients compared to healthy and latent TB individuals [59].

The relevance of IP-10 extends to TB treatment monitoring, with several studies indicating a decrease in IP-10 levels during treatment [60,61]. In a meta-analysis, IP-10 showed decreased levels by week 8 of treatment in comparison to the start of treatment [61]. IP-10 showed an average decrease of −38.2% (95% CI, −61.3% to −15.0%) [62,63,64,65] in this analysis [61]. However, heterogeneity in data reporting and follow-up timepoints underscores the need for standardised study design and reporting guidelines [61].

IP-10 levels decreased significantly after 6 weeks of treatment [66]. IP-10 levels have been correlated with TB treatment success in patients with extrapulmonary TB, with a significant reduction in IP-10 levels in 74% of good responders compared to 52% of partial responders [67]. Combination with information on anti-TB drug concentrations would be valuable, as for several anti-TB drugs, PK/PD targets have been proposed [68,69]. In a prospective study of patients with culture-confirmed drug-susceptible TB, AUC/MIC indices calculated for several anti-TB drugs were associated with clinical outcome and adverse effect prediction [70]. Similarly, in a prospective study of patients with multi-drug-resistant TB, both drug exposure (AUC) and MIC were associated with response to treatment. The investigators identified AUC/MIC thresholds from their results, but their utility has yet to be evaluated in prospective randomised controlled trials [71]. Both studies relied on culture conversion, which is difficult to use as there is a delay in obtaining test results in addition to limited test capacity in low-resource but high-burden settings. Hence, culture conversion does not seem to be a suitable PD parameter for the adjustment of dosages at the beginning of therapy. 

Monitoring both IP-10 as a PD biomarker in combination with anti-TB drug concentrations may be informative in aiding early dosing decisions. However, an understanding of the expected trajectory of IP-10 levels, especially immediately after the initiation of treatment, is required.

## 6. Bacterial Sepsis and Procalcitonin

Sepsis is a life-threatening complication of infection, usually bacterial infection, whereby the body has a severe inflammatory response. Considering the damage sepsis has on the body’s tissues and organs (i.e., organ dysfunction) and the high mortality rates (20–50%), rapid, optimised treatment is essential [72,73,74]. 

Procalcitonin (PCT) is a peptide precursor of the hormone calcitonin. Stimulation of inflammatory cytokines during bacterial infections correlates with the release of PCT from tissues [75]. A PCT concentration ≥0.25 µg/L may be suggestive of bacterial infection [76]. The pooled sensitivity and specificity of PCT for the diagnosis of sepsis have been shown to be 0.80 and 0.75, respectively [77]. However, this is not considered reliable to guide the initiation of antibiotics in patients with sepsis [78]. However, serial PCT measurements to guide ongoing antibiotic therapy in critically ill sepsis patients are associated with lower mortality and a shorter duration of antibiotic therapy [79]. Thus, once antibiotics have been initiated, there is some evidence to use PCT to support ongoing antibiotic decisions [78].

The use of PCT as an adjunct to antibiotic TDM may inform clinical decision making in specific situations. First, persistently elevated PCT in patients with therapeutic drug exposure may be indicative of treatment failure. Inadequate response to antibiotics in those with normal concentrations in the blood could indicate poor drug penetration at the site of infection. This is both site- and antibiotic-dependent. Second, a low drug concentration may not require a dose increase in patients with declining PCT, especially with clinical improvement.

In order to use PCT as an adjunct to TDM, an understanding of the expected trajectory of PCT during treatment and thresholds for decisions is required. Assuming adequate treatment in patients with sepsis, the half-life of PCT is 1–1.5 days and is not dependent on renal function [80]. A PCT <0.25–0.5 µg/L or decrease of >80% has been suggested as a threshold for antibiotic discontinuation [76]. While there is no evidence to define what rate of decline is too slow, no change or increase in PCT in the first 72 h is a poor predictor of survival [81]. Thus, if PCT at 72 h does not decline or increases, then a dose increase or antibiotic change should be considered, even if TDM indicates concentrations in the expected range. A randomised controlled trial showed that a change in antibiotic can be recommended if PCT increased from baseline peak concentrations and was ≥0.5 µg/L [82]. Alternatively, drug dosing does not need to be increased if PCT is declining and between 0.25 and 0.5 µg/L in the context of low antibiotic concentrations. This approach requires consideration of drug toxicity and the clinical improvement of the patient. For the use of PCT as an adjunct to TDM, the change in the rate of PCT decline is pertinent. One important pitfall is that PCT concentrations can decrease up to 85% after initiation of renal replacement therapy, and serial measurements may therefore not be useful to guide TDM in these patients [83].

## 7. Discussion

Optimising antimicrobial drug regimens using an integrated PK and PD approach is an emerging concept of interest in the field of infectious diseases. In this paper, we shared our perspective on the ability to use diagnostic biomarkers to monitor treatment response and, together with TDM, make more informed dosing decisions.

Biomarkers have an important role in the diagnosis of infectious diseases as they can be readily tested (usually using blood samples) and can be quantified immediately (within a few hours). Unlike traditional methods of diagnosis, like culture, biomarkers can therefore be used to rapidly diagnose diseases and initiate therapy sooner, which is critical in immunocompromised patients. Ensuring patients are on the correct drug at the right dose as quickly as possible is just as important as early detection of disease. While biomarkers have traditionally been used in diagnosis, they also have an emerging role in monitoring antimicrobial treatment responses. It is logical to interpret PD biomarker concentrations alongside TDM, which is often utilised to assess drug exposure in patients not adequately responding to therapy. 

By considering both biomarker response and drug concentrations together, a more informed decision can be made on whether dose adjustment or switching to an alternative therapy is required. The biomarker response (decrease, stable or increase) provides insight into the PD of the drug, while TDM shows the PK of the drug in an individual patient. Compared to current practice, where only the drug concentration is used to guide drug dosing decisions, the interpretation of a drug concentration in light of the biomarker response will be more informative (Figure 1).

Other fields like oncology and inflammatory conditions have already recognised the possibility, convenience and utility of using biomarkers to assess treatment response [84,85,86]. In oncology, several protein tumour markers have been identified in various tumour types and linked with responses to chemotherapy. Using these biomarkers to avoid inappropriate dose escalations and prolonged treatment has been identified as a potential benefit but not researched extensively yet [86,87]. In the treatment of Crohn’s disease, monitoring both faecal calprotectin, a diagnostic biomarker, and infliximab concentrations to guide dose adjustments was found to increase the chance of endoscopic response and remission [88]. An association has been found between infliximab exposure, biochemical remission and clinical remission [89]. In addition, persistent increases in faecal calprotectin concentrations despite dose escalations were associated with a lack of response and remission [88]. Biomarker research should also be pioneered in the infectious diseases field.

While there is not enough evidence to make a definitive statement on the utility of the combined biomarker-TDM approach, and different biomarkers are in different stages of research, we think that this approach has a broader application to the optimal use of antimicrobials. Indeed, biomarkers are already being used to guide the initiation and termination of antimicrobials for certain infections.

## 8. Future Directions

### 8.1. Next Steps

While specific infections and biomarkers were described in this paper, other diagnostic biomarkers such as soluble Triggering Receptor Expressed on myeloid cells-1 (sTREM-1) and presepsin are also available [90]. Before antimicrobial drug exposure (PK) and biomarker testing (PD) can become standards of care, both need to be well studied. More specifically, we need the following:To determine the most suitable exposure target. This involves defining the best PK parameter to monitor for and then the therapeutic drug target in terms of that parameter. While traditionally TDM involves the collection of trough concentrations, this parameter might not be a good measure of drug exposure for all antimicrobials, just like ganciclovir. The AUC or maximum concentration may be more suitable. Both preclinical and clinical cohort studies will be important for this step. Hollow fibre infection models are dynamic two-compartment in vitro models that allow the simulation of in vivo drug exposure [91]. PK data can be collected from these models while accounting for PD factors, making them suitable for preclinical studies [91,92].A better understanding of the trajectory of biomarker levels from baseline to immediately after the initiation of therapy to the end of therapy, including treatment failure and death, is needed. Cohort studies are needed to obtain these data, and they will likely involve frequent sample collection, at least initially. Designing studies to use left-over samples (e.g., from the ICU) may reduce the burden on participants. The half-life of the biomarker needs to be determined to identify how frequently it could be sampled to interpret changes in biomarker levels correctly. Once the data is available, it should be modelled to better understand the biomarker trajectories and facilitate sparse sampling.To determine if other factors such as renal function, hepatic function, inflammation and/or concomitant medications might impact biomarker readings [93]. Patient populations where biomarker testing has limited sensitivity and specificity need to be identified so biomarker readings are not overinterpreted and inappropriate antimicrobial dosing decisions are not made [31].Before clinical implementation, prospective studies are needed to confirm if using this approach is indeed more effective in optimising therapy compared to traditional TDM. The study design should focus on evaluating the effect of biomarker-/TDM-informed dosing on patient outcomes, including mortality, response and toxicity [94]. Additionally, based on steps 1–3, exposure targets; frequency and time of sampling; and confounding factors to consider should be pre-defined.Evaluation of toxicity biomarkers to further inform dosing decisions. In addition to treatment efficacy, drug safety is also vital in the individualisation of therapy. Certain biomarkers can be used to evaluate drug toxicity [95]. Taking hepatotoxicity as an example, biomarkers such as alanine aminotransferase, aspartate aminotransferase, alkaline phosphatase and other liver function tests are used to detect hepatotoxicity. By assessing changes in such biomarker values at the start of therapy and throughout, early signs of toxicity can be detected, and dose or treatment can be modified to prevent significant toxicity [96].

### 8.2. Biosensors

To further optimise the logistics of TDM and biomarker monitoring, biosensors could be applied, providing real-time data on drug concentrations and biomolecules. Having sensors available would significantly reduce study costs and reduce the burden on participants, as frequent sampling can be avoided [97]. The first-in-human real-time drug concentration monitoring of an antimicrobial (phenoxymethylpenicillin) has been attempted in healthy volunteers using a microneedle-based β-lactam biosensor (worn in the participants forearm). The PK profiles from the biosensor were similar to those generated from microdialysis, suggesting feasibility for wider application in clinical practice [98]. As for biomarkers, biosensors have been developed for the detection of galactomannan, CMV viral load, BDG, IP-10 and PCT [99,100,101,102,103]. However, all need to be researched further in animals and humans before attempting implementation in clinical practice [97,102]. 

### 8.3. Building on Current Knowledge

While traditional trough-level guided proportional dose adjustments are being replaced with model-informed precision dosing to determine the best possible dose for a patient using patient characteristics, the next step will be to include biomarker responses [104]. This would help to move away from indices like AUC/MIC and advance the PK/PD mathematical models that are being developed [105]. A novel index of AUC for the concentration of antimicrobials that induces a half-maximal reduction in biomarker levels could also be used. This has already been attempted with antifungals and galactomannan, but with a small dataset [23]. Larger datasets for each antimicrobial biomarker should be modelled to determine if this index can be utilised.

## Figures and Tables

**Figure 1 pharmaceutics-16-00677-f001:**
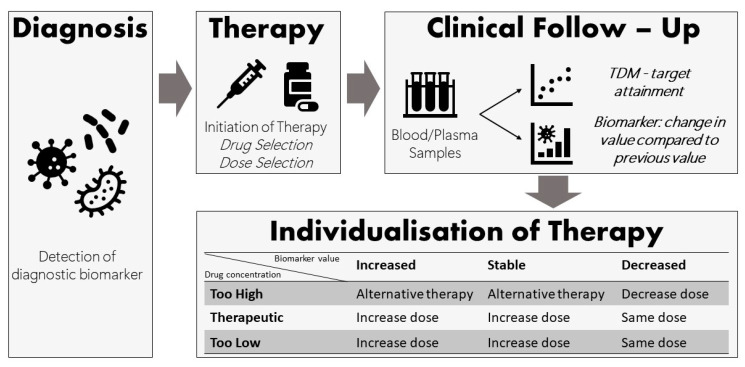
Utilisation of both TDM data and biomarker responses to guide antimicrobial dosing decisions and individualise therapy.

## Data Availability

No new data were created or analysed in this study. Data sharing is not applicable to this article.

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
