# Peer review of "Therapeutic Drug Monitoring and Biomarkers; towards Better Dosing of Antimicrobial Therapy"

_pharmaceutics, 2024, doi:10.3390/pharmaceutics16050677_

Round 1

Reviewer 1 Report

Comments and Suggestions for Authors

The presented work reviews the current knowledge on the use of biomarkers as pharmacodynamic indices in PK/PD modeling and antibiotic dosage optimization. In times of increasing resistance to antibiotics and a small number of new antimicrobial drugs appearing on the pharmaceutical market, optimizing dosage to achieve maximum therapeutic effect seems to be the best strategy.

I have no comments about the work.

Author Response

Thanks for reviewing the manuscript.

Reviewer 2 Report

Comments and Suggestions for Authors

A review article "Role of biomarkers in the management of antibiotic therapy: an expert panel review: I – currently available biomarkers for clinical use in acute infections" and many other such articles are available. Why the authors have not used this review? Is it because the review is about non-specific biomarkers?

1.       The author after introduction has focused only on a few antimicrobials and their correlation to the biomarkers. The authors have not discussed/mentioned (referred to) the following biomarkers; Myeloid cells-1 (sTREM-1), Soluble urokinase-type Plasminogen receptor (suPAR), proadrenomedullin (ProADM), and Presepsin etc. Even the authors have used one reference regarding Presepsin. 

2.   Related to the above biomarkers, immune response can also be used as biomarkers.

3. The national institute has defined the subtypes of biomarkers. The authors have not mentioned which type of biomarker(s) they are referring to? If they are referring to any specific one then reason must be given?

4.  The authors described that in some cases plasma might not reflect intracellular levels. Other fluids such as CSF, BAL, and pleural fluids can be analyzed for TDM. Can the biomarkers be identified in other fluids?

5. Analyzing both drug concentration and biomarker will definitely increase the treatment cost. Is there any possibility of avoiding TDM and relying only on biomarkers? Give justification

6.       The authors should submit a neat format of the manuscript. This seems to be the manuscript which they have internally reviewed with comments in the review box.

7.       Hollow fibre infection models described in future directions is not something related to biomarkers study. It is used for dose optimization and to overcome antibiotic resistance.

8.       The authors may also include the biomarkers related to the toxicity of the antibiotics.

Comments on the Quality of English Language

Minor changes required.

Reviewer 3 Report

Comments and Suggestions for Authors

The paper by Wehbe et al. deals with the important issue of the use of biomarkers in the TDM guided process of drug therapy optimization.

The paper has therefore great potential, however, in its present form it has many qualitative limitations.

In general, the paper is in disarray and concepts are not presented in an organic way. Concepts in Discussion should find a better placement in Introduction, since these are known general gaps in knowledge, not specific issues related to the present paper.

Introduction starts (first phrase) unexpectedly with immunocompromised patients.. We would suggest a broad view and an eventual focus, instead. Anyway, immunocompromised patients are a particular and difficult to treat population that deserve, eventually, a separate and thorough in-depth analysis.

Successively, five biomarker examples are examined separately: i.e. galactomannan, BDG,viral CMV load, IP-10 and PCT. 

Each marker has specific limitations that authors correctly describe. However, it is unclear in the end if any of these could be used and how. We suggest elaborate diagrams of workflows depicting the exact strategies and eventual limitations on how they propose to use each biomarker.

These can be used as a guide for Discussion.

A Discussion on biomarkers in general, as the current one, is generic and misguiding.

Every specific marker additionally has analytical limitations that have not been even issued, let alone specifically addressed.

We think that analytical limitations of biomarker assessment should be considered.

In TDM, assessing drug efficacy levels through biomarkers in one part of the story, the other part is evaluating toxicity.

We suggest they also  mention useful biomarkers for toxicity. Information can be found in reviews in the literature, such as https://doi.org/10.1093/nar/gkad862

Comments on the Quality of English Language

English in need of professional revision

Author Response

.

Round 2

Reviewer 3 Report

Comments and Suggestions for Authors

Authors have specifically addressed the issues we previously raised. The concept of biomarker based TDM can be a valuable one in the clinics and we therefore suggest the paper is considered for publication.

In addition , the explicative diagram presented in the new version can be a valuable item that should increase the appeal of the paper.

Discussion has been improved with the addition of comments and study limitations.